# Spatial–Spectral Evidence of Glare Influence on Hyperspectral Acquisitions

**DOI:** 10.3390/s20164374

**Published:** 2020-08-05

**Authors:** Alberto Signoroni, Mauro Conte, Alice Plutino, Alessandro Rizzi

**Affiliations:** 1Information Engineering Department, Università degli Studi di Brescia, via Branze 38, I-25123 Brescia, Italy; alberto.signoroni@unibs.it (A.S.); m.conte@studenti.unibs.it (M.C.); 2Computer Science Department, Università degli Studi di Milano, via Celoria 18, 20133 Milan, Italy; alice.plutino@unimi.it

**Keywords:** glare, hyperspectral imaging, digital imaging

## Abstract

Glare is an unwanted optical phenomenon which affects imaging systems with optics. This paper presents for the first time a set of hyperspectral image (HSI) acquisitions and measurements to verify how glare affects acquired HSI data in standard conditions. We acquired two ColorCheckers (CCs) in three different lighting conditions, with different backgrounds, different exposure times, and different orientations. The reflectance spectra obtained from the imaging system have been compared to pointwise reference measures obtained with contact spectrophotometers. To assess and identify the influence of glare, we present the Glare Effect (GE) index, which compares the contrast of the grayscale patches of the CC in the hyperspectral images with the contrast of the reference spectra of the same patches. We evaluate, in both spatial and spectral domains, the amount of glare affecting every hyperspectral image in each acquisition scenario, clearly evidencing an unwanted light contribution to the reflectance spectra of each point, which increases especially for darker pixels and pixels close to light sources or bright patches.

## 1. Introduction

Glare, also referred to as flare, is often defined as a Human Visual System (HSV) condition which causes a reduced capability to see details, due to an unsuitable distribution of luminance in the scene [1,2]. Unfortunately, the term “glare” is used to refer both to the cause and to the effect of this phenomenon, as in the definition above which is a description of the effects, not of the cause. In the HVS, glare is caused by the scattering that occurs in the eye-bulb that spreads a fraction of light across the retina reducing the projected contrast, usually causing a vision loss. It has been found that intra-ocular glare and cortical processing change the appearance of the observed scene, that rarely correlates with the luminance coming from it [3,4]. Furthermore, glare affects the perceived details and dynamic range [5] and causes discomfort [6], usually in scenes with high dynamic range [7]. Here we want to address a very different effect of glare: how it affects systematically the acquisition of an image, in this case hyperspectral. In fact, this phenomenon is not just a characteristic of human vision, it affects every image acquisition system that uses a lens. Generally, it causes a scene and camera dependent unwanted scattering of light spread across the image sensors [8] (see Figure 1). The glare problem is well known in the field of lens design optics, but much less studied and considered in the imaging field, where measures and tests on glare in image acquisition systems started just some years ago [9]. It these studies, it has been found that glare causes unwanted changes of pixel values and limits the radiance range which can be acquired in a shot. A more extensive explanation of glare effects in standard and HDR images acquisition can be found in [10] and in [11], while there is still a lack of studies in the field of spectral imaging, and this seems to contrast with the fact that a hyperspectral image HSI is more oriented towards measurement than perception. Moreover, HSI is becoming increasingly popular in many application fields, especially where added value is generated by the combination of digital imaging and spectroscopy [12,13,14,15,16].

To achieve a trustworthy qualitative and quantitative analysis of HSI data, it is fundamental to calibrate the system, compensate for the distorting effects, and obtain comparable reflectance spectra at intra- and inter-data level. HSI acquisitions are affected by a number of issues—such as uneven receptor sensitivity, dark current, glare, interference pattern, non-uniform lighting, atmospheric and topographic effects, parasitic images, reflections and shadows—that require more or less sophisticated compensations, calibrations or removal actions. Unfortunately, glare is not just noise. First, its effect is relevant, and glare interference can weight much more than well known and better studied noise sources. Second, it is a systematic, unavoidable and, de facto, unpredictable increment of pixel values, dependent on the spatial distribution of radiance in the acquired scene. An attempt to remove glare in image acquisition was made by Tavlata et al. in [17] using an occlusion mask. A simplification of the work of Tavlata et al. can be found in [5]. However, the need for a mask makes the acquisitions unpractical. The authors of [18] discussed the main reasons why glare noise can not be easily removed in the image acquisition process. For each step of glare formation, a corresponding influence is analyzed in terms of ill-posedness and ill-conditioning of the problem. In this contribution, we aim at characterizing the presence of glare in HSI acquisitions, at measuring its contribution in the final image and at characterizing its spectral variations. To this end, we compare the contrast of the grayscale patches of a ColorChecker in standard colorimetric conditions with the ones obtained through HSI imaging, to identify the variations introduced by the optics and so the contribution imputable to glare in the final image (Figure 1). Our experiments aim to increase the awareness of glare and define how it can affect HSI measurements. Eventually we discuss the differences between discomfort glare or specular non-uniformity in imaging (two particular cases that happen in certain conditions) compared to the systematic effect of lens glare that occurs in every acquisition, also under uniform diffused illumination [6,7].

## 2. Materials

### 2.1. Hyperspectral Camera and Calibration

The hyperspectral images of this study have been taken with a SpecimIQ camera (https://www.specim.fi/iq/), which acquires data in the VNIR wavelength range from 400 nm to 1000 nm with a spectral resolution of 7 nm.

SpecimIQ is an handheld camera that acquires HSI data according to classical dispersive push-broom principles, i.e., spatial–spectral temporal frames (x−λ dimensions) are acquired in a space-varying sequence (in the *y* dimension). One peculiarity of the SpecimIQ camera is the presence of an internal motorized opto-mechanical moving system that enables scanning without the need for relative motion between the device and the sample (or scene). This gives a user experience similar to the one of shooting a scene with a normal camera mounted on a tripod, except for the need that the object/scene remains still for the needed acquisition time (from few to tens of seconds, depending on exposure time and light conditions balance). A service RGB camera helps the user in composing the frame (even if the FOV is not perfectly matched with the one of HSI optics, especially for closer shooting distances). Dedicated camera calibration software allows practical and intuitive acquisitions directly controlled by the user though the device touch screen. On the same screen, quick true-color visualization of the acquired data cube is provided by the camera software combining three reference wavelengths: λ= 598 nm (R channel), λ= 548 nm (G channel) and λ= 449 nm (B channel).

Hereafter, we recall the parameters and calibration procedure of main interest for this study, while for an in-depth overview of the camera, the reader can refer to the manufacturer’s website and to the camera evaluation work [19]. We represent the acquired data cube as
(1)$$\begin{array}{c} HSI=X\times Y\times L\to R\\ x,y,l\mapsto HSI(x,y,l) \end{array}$$
where the maximum available range of spatial/spectral pixels/bands, i.e., without spatial or spectral binning, are:(2)$$\begin{array}{c} X=Y=\{0,\cdots ,511\}\\ \Lambda =\{0,\cdots ,203\} \end{array}$$

Associated to the set *L* is the set of the corresponding wavelength $\Lambda =\{\lambda_{0},\lambda_{1},\cdots ,\lambda_{203}\}$ that spans the 400–1000 nm VNIR range.

Absolute reflectance measures can be obtained by normalization of the acquired data. In our setup, we can easily acquire three different data: the acquired raw data-cube $R_{DC}$, a black (dark current) frame $D_{F}$ and white reference spectrum $W_{R}$. The $W_{R}$ spectrum is preferably associated with each acquisition and derived from averaging the pixel in an area corresponding to a (usually white) calibration target placed in the scene or near the object of interest before the acquisition. With the white target (usually made of an opto-polymer with reflectance >99% over the whole wavelength range of interest) we are able to measure and compensate from the spectral non-uniformity of the illuminating source. The above data are used by the camera software that produces the final pixel-wise wavelength-dependent reflectance values R(x,y,l) according to the following calibration formula:(3)$$R\left(x,y,l\right)=\frac{R_{DC}\left(x,y,l\right)-D_{F}\left(x,l\right)}{W_{R}\left(l\right)-D_{F}\left(x,l\right)}$$

This way we are able to produce reflectance data we can use and compare across different acquisitions since they are compensated from the main biases. This also allows to decouple glare (not at all compensated this way) from other distortions. However, in HSI acquisition under natural or artificial light it is hard to presume or impose the spatial homogeneity of lighting. As consequence, the acquisitions are affected by effects resulting from this non-homogeneity and, since the white calibration is made through the average of the spectrum in a reduced area, the non-homogeneity is not compensated by Equation (Equation 3). This affects measures that aim, as in our case, to compare reflectances coming from different areas of the scene. We will take into account these issues in Section 4.1.

### 2.2. ColorCheckers and Reference Spectra

In this work, we focus our analysis on the grayscale patches of the ColorChecker and we will refer to these according to the official notation in the following way (Table 1):

Usually ColorCheckers (CCs) come with associated reference spectral data in the visible range (380 nm to 730 nm). They are typically provided at uniform intervals of 10 nm [20]. In this paper, these tabulated reference data, here named $RS^{\left(1\right)}$, are plotted for every level in the black-white scale with dashed lines. The standard reference $RS^{\left(1\right)}$ is used to make comparisons and to compute the glare effect in the visible range (from 400 to 700 nm).

However, since our spectral range of interest is wider (400 nm to 1000 nm), we also acquired a second set $RS^{\left(2\right)}$ of reference reflectance spectra. These data were acquired using an Ocean Optics spectrophotometer (HR4000), with spectral sensitivity range from 200 nm to 1000 nm and average spectral resolution of 0.24
nm. The acquisition angle was fixed at 45° and a Y-branched QR400-7-VIS-NIR optical fibre bundle was used for the acquisition. $RS^{\left(2\right)}$ curves for each gray level are plotted with a dotted line. Since the measures are acquired with a different instrument and with a different light source the measures present a mismatch visible in the figures of Section 3.1. Due to this we decided to use this as reference just in the IR region from 700 to 1000 nm, for the purpose of limiting the errors propagation.

The reference spectra can be formalized as:(4)$$\begin{array}{c} RS_{c}^{\left(i\right)}:L^{\left(i\right)}\to R\\ l\mapsto RS_{c}^{\left(i\right)}\left(l\right) \end{array}$$
where,
(5)$$\begin{array}{c} L^{\left(1\right)}=\left\{0\cdots 33\right\}\\ L^{\left(2\right)}=\left\{0\cdots 3645\right\} \end{array}$$

As for the data acquired from the camera, $L^{\left(1\right)}$ and $L^{\left(2\right)}$ are associated to two set of wavelength $\Lambda^{\left(1\right)}=\left\{\lambda_{0}^{\left(1\right)},\lambda_{1}^{\left(1\right)},\cdots \lambda_{3}^{\left(1\right)}\right\}$ and $\Lambda^{\left(2\right)}=\left\{\lambda_{0}^{\left(2\right)},\lambda_{1}^{\left(2\right)},\cdots \lambda_{3}^{\left(2\right)}\right\}$ ranging respectively 400 nm to 730 nm for the visible range and 700 to 1000 nm for the IR range.

### 2.3. HSI Data Acquisition Setups

With the hyperspectral camera, we acquired 23 hyperspectral images named from *01* to *23*, of two Gretag Macbeth ColorCheckers (http://poynton.ca/notes/color/GretagMacbeth-ColorChecker.html) (CC). All the images were acquired in the same room with the same artificial illumination (normal room neon lamps +2 AC powered 500 W halogen spotlights) in three different setups with some sub-configurations each (see Figure 2 and Figure 3 and Table 2, Table 3 and Table 4):Setup 1 (images from 01 to 14): the two ColorCheckers were placed on the same plane. Images from 03 to 08 were acquired with a black cardboard sheet as a background, the others with a white one.Setup 2 (images from 15 to 23): the two CCs were placed on two different planes (at about 120 cm of distance) with an LED lamp between them placed on an upper plane (closer to the camera) with a concave mirror below the lamp preventing backward lighting toward the CCs. All the images were acquired with a black cardboard sheet as a background.Setup 3 (images from 22 and 23): same as Setup 2 but without the LED lamp.

We will refer to the ColorCheckers in the same image with the name CCA for the first CC from left to right or from top to bottom in the image and CCB for the opposite one. In every image, we can find CCs with the gray-scale sector oriented towards the center of the image or, conversely, flipped towards the borders. We refer to the one or the other situation with ‘D0’ and ‘D1’ respectively. Thus, in the ‘D0’ condition, CCA has the grayscale with the ‘w’ patch near the border of the image and the ‘bk’ patch near the white reference ($W_{R}$), at the opposite CCB has the ‘bk’ patch near the borders of the image and the ‘w’ patch near $W_{R}$ (Figure 3a,c). In ‘D1’ condition the grayscale of CCA and CCB point in the opposite direction and are oriented toward the borders of the image (Figure 3b,d). To distinguish the background color we use ‘B0’ for black (Figure 3a) and ‘B1’ for white (Figure 3b). The single frame (in the push-broom sequence) exposure time, expressed in ms, is indicated at the end of the image name which is coded using the naming pattern N{·}S{·}B{·}D{·}-T{·}. For example image ‘01’, being in the first setup with a black background and grayscale towards the center and exposition 41 ms, will be called “N01S1B0D0-T41”. In Table 2, Table 3 and Table 4 are reported the naming of all the acquisition for each setup.

## 3. Methods

### 3.1. Spectral–Spatial Measures

Spectral–spatial behaviour was investigated by extracting spectra from different areas of every grayscale patch of the CCs. We extracted spectra coming from five different regions: one around the center and four near the corners. The five selected regions did not have fixed positions and dimensions as shown in Figure 4. For each region in the patch, we computed a spectrum by spatially averaging values at each wavelength. This way, each spectra could be considered a function assigning values to each wavelength:(6)$$\begin{array}{c} s_{c}:\Lambda \to R\\ l\mapsto s_{c}\left(l\right) \end{array}$$
where $s_{c}$ is the spectrum of the color patch *c* (e.g., *w, g3, g2, g1, g0, bk*).

To take into account the effect of illumination we also collected spectra information coming from two background regions in each image near the gray-scale of the CC, as shown in Figure 5 and Figure 6.

For each considered gray colored patch *c* in each image it was possible to plot five spectra (i.e., computed from the five different regions) and if the plot was about reflectance we could show the two reference spectra, one from the ColorChecker $rs_{c}^{\left(1\right)}$ and the other from the spectrophotometer $rs_{c}^{\left(2\right)}$. Example of spectral acquisitions on the CC of image N09 are illustrated in Figure 7, where Figure 7a shows reflectance data while Figure 7b shows raw data.

### 3.2. Glare Assessment

To measure the Glare Evidence (GE) in the Hyperspectral acquisitions we compared the contrast defined on the grayscale patches of the CC between the references and the HSI measures. We supposed that the main glare contribution was given by the proximity to bright areas (in all setups) or to light sources in the scene (in the second set up). As a consequence, the darker patches should be more affected by glare than the lighter ones.

To measure this phenomenon we computed the ratio between the spectrum of a gray level patch and the spectrum of the white patch of the same CC. Then, this first ratio was compared with the ratio of the spectrum of the same gray patch and the white patch in the reference CC (see Figure 8). In the presence of glare, the ratio of the two measured spectra was higher than the ratio of the two reference spectra.

Specifically, for a given sequence of gray patches in a CC we started considering all the ratios between each gray patch spectrum, $s_{c}$ and the spectrum of the white in that sequence, $s_{w}$.
(7)$$\begin{array}{c} r_{c,w}\left(l\right)=\frac{s_{c}\left(l\right)}{s_{w}\left(l\right)} \end{array}$$

To obtain a spectrum at specific wavelength position *l*, an operation of linear interpolation was performed. If ${\hat{\lambda }}_{l_{0}}$ and ${\hat{\lambda }}_{l_{1}}$ were the nearest wavelengths to a needed $\hat{\lambda }$ and the corresponding values were $s(l_{0})$ and $s(l_{1})$:(8)$$s\left(\hat{\lambda }\right)=\frac{1}{|{\hat{\lambda }}_{l_{0}}-{\hat{\lambda }}_{l_{1}}|}\left(\right|\hat{\lambda }-{\hat{\lambda }}_{l_{1}}|s\left(l_{0}\right)+|\hat{\lambda }-{\hat{\lambda }}_{l_{0}}\left|s\left(l_{1}\right)\right)$$

To get a compact representation of the information given by the above spectral ratios and to better highlight the presence of glare, we introduced a Glare Evidence (GE) parameter:(9)$$GE_{\lambda_{min},\lambda_{max}}^{\left(j\right)}\left(c\right)=\frac{f_{\lambda_{min},\lambda_{max}}\left(s_{c},s_{w}\right)}{f_{\lambda_{min},\lambda_{max}}\left(RS_{c}^{\left(j\right)},RS_{w}^{\left(j\right)}\right)}$$
where *c* refers to one of the grayscale patches *w, g3, g2, g1, g0* and *bk*, $\lambda_{min}$ and $\lambda_{max}$ specifies the range of wavelengths, or a single wavelength ($\lambda_{min}=\lambda_{max}$) and j=(1,2) refers to the reference spectra data-set. Numerator and denominator were both ratios between one of the gray spectra and the white spectrum, the former took into account values acquired by the camera, the latter uses values from one of the two reference spectra. For the first considered point, GE was equal to one by construction and, due to the presence of glare, we expected that this function increased from the white point (*w*) to the black (*bk*) because darker colors were much more affected by this phenomenon.

The reference spectra were pointwise measures, which were not subject to the interference of neighboring regions, unstable distribution of radiance, or light scattering on the sensors, so they were exempt from glare noise. Thanks to this, the ratio between the white and grayscale spectra in the HSI and in the references could provide us a trustworthy measure of the glare introduced by the camera. In fact, if the pointwise and HSI systems were equivalent, the ratio would have been one everywhere.

The function $f_{\lambda_{1},\lambda_{2}}$ depending on $\lambda_{1}$,$\lambda_{2}$ can take into account spectra values for a single wavelength value or for a range of wavelength.
(10)$$f_{\lambda_{min},\lambda_{max}}\left(a,b\right)=\left\{\begin{array}{cc} \frac{a(l)}{b(l)} & \lambda_{min}=\lambda_{max}\\ \;\\ \frac{1}{|L_{\lambda_{min},\lambda_{max}}|}\sum_{l\in L_{\lambda_{min},\lambda_{max}}}\frac{a(l)}{b(l)} & \lambda_{min}\neq \lambda_{max} \end{array}\right.$$
with
(11)$$L_{\lambda_{min},\lambda_{max}}=\left\{l\in L^{*}:\lambda_{l}\in \Lambda^{*},\lambda_{min}\leq \lambda_{l}\leq \lambda_{max}\right\}\;$$
where $L^{*}$ is the set of wavelengths on which the spectra *a* and *b* are defined (*L* for data acquired by the camera, $L^{1}$ for $RS^{1}$, $L^{2}$ for $RS^{2}$).

This way, it was possible to produce different plots combining the acquired data with the reference spectra ($RS^{\left(1\right)}$, $RS^{\left(2\right)}$), in a defined spectral range, or for each wavelength. For example, Figure 9a represents GE values for the various CC patches in the full visible range from 400 nm to 700 nm where different columns with same color refers to different measured patch areas. The curves in Figure 9b are spectral trends where in particular GE curves were quite smooth thanks to a sliding window averaging.

## 4. Results and Discussion

For the values of Glare Evidence, as explained in the previous section, we expected increasing values going from ‘w’ to ‘bk’ with a minimum value of 1 for the white patch ‘w’ and the highest value for the black patch ‘bk’. The results of the first setup (‘S1’) were analyzed and discussed in Section 4.1 and reported in Table 5 and Table 6, where $GE_{400,700}^{\left(1\right)}$ and $GE_{700,1000}^{\left(2\right)}$ are considered respectively for the images with exposure time of 41 ms. In this section, in Table 7 the results obtained through different exposure times are analyzed. The results of the second (‘S2’) and third setup (‘S3’) are discussed in Section 4.2. The results of the second setup are reported in Table 8 and Table 9, where are considered $GE_{400,700}^{\left(1\right)}$ and $GE_{700,1000}^{\left(2\right)}$ for images with exposure time of 44, 42 and 11 ms. The GE values in the third setup are computed from images acquired with exposure time of 21 and 11 ms, and the results are reported in Table 10 and Table 11. In Section 4.3 we provide results and considerations about spectral GE behaviors.

### 4.1. Setup 1

In this section the comparison between the reference reflectance values of the ColorChecker and the reflectance measured through HSI are reported. Here we focus on the first setup (‘S1’), where the CCs have been acquired with a black (‘B0’) or white (‘B1’) background, with an exposure time of 41 ms (‘T41’). The Glare Effect in the visible range ($GE_{400,700}^{\left(1\right)}$) has been measured using as reference $RS^{1}$:

**Table 5 sensors-20-04374-t005:** $GE_{400,700}^{\left(1\right)}$.

HSI Number	w	g3	g2	g1	g0	bk
N01S1B0D0-T41-CCA	1.00	0.99	1.01	0.98	1.02	1.29
N01S1B0D0-T41-CCB	1.00	1.06	1.10	1.13	1.27	1.56
N02S1B0D1-T41-CCA	1.00	1.04	1.08	1.10	1.22	1.52
N02S1B0D1-T41-CCB	1.00	1.03	1.01	0.98	1.05	1.28
N07S1B1D1-T41-CCA	1.00	1.04	1.11	1.18	1.42	2.22
N07S1B1D1-T41-CCB	1.00	1.03	1.04	1.06	1.25	1.90
N08S1B1D0-T41-CCA	1.00	1.00	1.03	1.04	1.18	1.80
N08S1B1D0-T41-CCB	1.00	1.07	1.12	1.19	1.43	2.14

To make the table (of the glare effect in each acquisition) easier to read we highlighted in gray the values between 1 and 1.19, in light blue the values between 1.20 and 1.79, yellow the values between 1.80 and 2.39.

The Glare Effect in the IR range ($GE_{700,1000}^{\left(2\right)}$) has been measured using as reference $RS^{2}$:

**Table 6 sensors-20-04374-t006:** $GE_{700,1000}^{\left(2\right)}$.

HSI Number	w	g3	g2	g1	g0	bk
N01S1B0D0-T41-CCA	1.00	0.87	0.81	0.87	0.95	1.55
N01S1B0D0-T41-CCB	1.00	0.96	0.91	1.08	1.23	2.02
N02S1B0D1-T41-CCA	1.00	0.92	0.90	1.02	1.20	2.01
N02S1B0D1-T41-CCB	1.00	0.91	0.82	0.91	0.98	1.58
N07S1B1D1-T41-CCA	1.00	0.94	0.95	1.16	1.49	2.98
N07S1B1D1-T41-CCB	1.00	0.93	0.87	1.05	1.28	2.49
N08S1B1D0-T41-CCA	1.00	0.88	0.85	0.97	1.17	2.27
N08S1B1D0-T41-CCB	1.00	0.97	0.95	1.19	1.46	2.82

To make the table (of the glare effect in each acquisition) easier to read we highlighted in gray the values between 1 and 1.19, in light blue the values between 1.20 and 1.79, yellow the values between 1.80 and 2.39 and in orange the values over 2.40.

The GE values in the visible range, reported in Table 5, increased as expected along the grayscale and the blacks are the patches with highest values, with the only exception of the CCB in ‘N02’. In this setup, it was possible to notice that the presence of glare in the images grew considerably when the black patches were near the white reference or in the condition with the white background.

We remind to the reader that the images were acquired with a black (‘B0’) or a white (‘B1’) background, and that the CCs could be displayed horizontally with both the grayscales toward the center of the image (‘D0’) or toward the borders of the image (‘D1’). Thus, looking at the GE values of Table 5 we can notice that in ‘B1’ conditions the values of ‘g3’, ‘g2’ and ‘g1’ had similar values than in condition ‘B0’, but the values of ‘g0’ and ‘bk’ were considerably higher.

Furthermore, we can notice that ‘N07-CCA’ and ‘N08-CCB’ presented the black patches with the highest GE values. In this case, both the CC were displayed on a white background and with the black patches toward the left border of the image. It can be noticed that the black patches in this area of the image, presented the highest values of GE. This effect could be caused by an irregular light distribution, which was more intense on the left side of the image. This condition contributed to producing a higher glare, especially with a white background which caused in ‘N07-CCA’ and ‘N08-CCB’ the highest values of GE. Notwithstanding the influence of the light distribution in the left region, it is fundamental to notice that this effect just increased the glare, which however affected all the measured black patches, also in the CC with ‘bk’ patch oriented toward the opposite direction.

In the IR range, reported in Table 6, the trend of the data was similar to the visible range. In the IR range, the incongruity and mismatch problems of the reference spectra were more visible, which caused the decrease of GE values under 1. Despite this problem, also in the IR range the black patch presented considerably higher values of GE, which in most cases grew over 2 (meaning over 100% values alteration). Also in this range of wavelengths, ‘N07-CCA’ and ‘N08-CCB’ presented the highest GE values and the data trend was the same as in the visible range.

In Table 5 and Table 6 we considered an exposure time of 41 ms, which was empirically defined as the more suitable for the setup 1. To support this decision we report here the GE values obtained acquiring an image in the same conditions of setup, background and direction, through different exposure times (‘T41’, ‘T33’, ‘T14’):

**Table 7 sensors-20-04374-t007:** $GE_{400,700}^{\left(1\right)}$.

HSI Number	w	g3	g2	g1	g0	bk
N01S1B0D0-T41-CCA	1.00	0.99	1.01	0.98	1.02	1.29
N01S1B0D0-T41-CCB	1.00	1.06	1.10	1.13	1.27	1.56
N03S1B0D0-T33-CCA	1.00	0.99	1.00	0.97	1.02	1.28
N03S1B0D0-T33-CCB	1.00	1.06	1.10	1.12	1.26	1.58
N05S1B0D0-T14-CCA	1.00	0.99	1.00	0.97	1.03	1.30
N05S1B0D0-T14-CCB	1.00	1.05	1.09	1.12	1.27	1.57

To make the table (of the glare effect in each acquisition) easier to read we highlighted in gray the values between 1 and 1.19, in light blue the values between 1.20 and 1.79.

The GE values in Table 7 showed that changing the exposure time, the results were nearly unchanged, with an average variance of 0.02.

### 4.2. Setup 2 and 3

In this section are reported the Glare Effect values for the setup 2 (‘S2’) and the setup 3 (‘S3’). In this case, all the hyperspectral images have been acquired with a black background (‘B0’) and the ColorCheckers have been disposed vertically over the two main directions ‘D0’ and ‘D1’. We remind to the reader that in setup 2 the acquisitions have been made with an LED lamp between the two CCs and in setup 3 without the lamp. Considering the setup 2, the Glare Effect in the visible range ($GE_{400,700}^{\left(1\right)}$) has been measured using as reference $RS^{1}$:

**Table 8 sensors-20-04374-t008:** $GE_{400,700}^{\left(1\right)}$.

HSI Number	w	g3	g2	g1	g0	bk
N13S2B0D1-T44-CCA	—	—	—	—	—	—
N13S2B0D1-T44-CCB	—	—	—	—	—	—
N14S2B0D0-T42-CCA	—	—	—	—	—	—
N14S2B0D0-T42-CCB	1.00	1.19	1.78	1.54	1.86	2.37
N15S2B0D0-T42-CCA	—	—	—	—	—	—
N15S2B0D0-T42-CCB	—	—	—	—	—	—
N16S2B0D0-T42-CCA	—	—	—	—	—	—
N16S2B0D0-T42-CCB	—	—	—	—	—	—
N17S2B0D0-T11-CCA	1.00	1.05	1.07	1.63	1.29	1.76
N17S2B0D0-T11-CCB	1.00	0.96	1.57	1.44	1.82	2.37
N18S2B0D0-T11-CCA	1.00	1.07	1.12	1.59	1.60	2.61
N18S2B0D0-T11-CCB	1.00	0.99	1.57	1.63	2.24	3.48
N19S2B0D1-T11-CCA	1.00	1.05	1.17	1.15	1.25	1.59
N19S2B0D1-T11-CCB	1.00	1.08	1.16	1.36	1.30	1.70
N20S2B0D1-T11-CCA	1.00	1.03	1.13	1.09	1.19	1.55
N20S2B0D1-T11-CCB	1.00	1.07	1.15	1.29	1.29	1.65
N21S2B0D1-T11-CCA	1.00	1.03	1.07	1.09	1.18	1.55
N21S2B0D1-T11-CCB	1.00	1.06	1.12	1.15	1.22	1.59

To make the table (of the glare effect in each acquisition) easier to read we highlighted in gray the values between 1 and 1.19, in light blue the values between 1.20 and 1.79, yellow the values between 1.80 and 2.39 and in orange the values over 2.40.

and the Glare Effect in the IR range ($GE_{700,1000}^{\left(2\right)}$) has been measured using as reference $RS^{2}$:

**Table 9 sensors-20-04374-t009:** $GE_{700,1000}^{\left(2\right)}$

HSI Number	w	g3	g2	g1	g0	bk
N13S2B0D1-T44-CCA	—	—	—	—	—	—
N13S2B0D1-T44-CCB	—	—	—	—	—	—
N14S2B0D0-T42-CCA	—	—	—	—	—	—
N14S2B0D0-T42-CCB	1.00	1.08	1.89	1.56	1.89	2.91
N15S2B0D0-T42-CCA	—	—	—	—	—	—
N15S2B0D0-T42-CCB	—	—	—	—	—	—
N16S2B0D0-T42-CCA	—	—	—	—	—	—
N16S2B0D0-T42-CCB	—	—	—	—	—	—
N17S2B0D0-T11-CCA	1.00	0.96	0.92	2.01	1.35	2.33
N17S2B0D0-T11-CCB	1.00	0.86	1.74	1.47	1.89	2.90
N18S2B0D0-T11-CCA	1.00	1.01	1.08	2.05	2.31	5.05
N18S2B0D0-T11-CCB	1.00	0.96	1.71	1.98	3.11	6.34
N19S2B0D1-T11-CCA	1.00	0.96	1.05	1.18	1.28	2.01
N19S2B0D1-T11-CCB	1.00	0.95	0.97	1.41	1.30	2.12
N20S2B0D1-T11-CCA	1.00	0.94	0.97	1.09	1.20	1.99
N20S2B0D1-T11-CCB	1.00	0.95	0.95	1.24	1.25	2.00
N21S2B0D1-T11-CCA	1.00	0.93	0.91	1.07	1.19	1.99
N21S2B0D1-T11-CCB	1.00	0.94	0.93	1.05	1.16	1.91

To make the table (of the glare effect in each acquisition) easier to read we highlighted in gray the values between 1 and 1.19, in light blue the values between 1.20 and 1.79, yellow the values between 1.80 and 2.39 and in orange the values over 2.40.

As reported in Table 8 and Table 9 in the second setup have been used different exposure times. In fact, the presence of an LED lamp between the two CC caused the saturation of the camera sensors for exposure times over 11, with just one exception (‘N14-CCB’).

Additionally in this setup, as in ‘S1’ presented in the previous section, all the GE values increased along the grayscale, and the black patch presented the highest values in all the CCs. In this setup, the main differences were caused by the changes in the orientations of the CCs. In fact, in the condition ‘D0’, where both the grayscales of ColorChecker A and B were near the LED lamp, we had the highest values of GE (i.e., ‘N17’ and ‘N18’) and the highest values of glare were registered for the ‘bk’ patches of ‘CCA’ and ‘CCB’ in the image ‘N18’. Thus, the proximity to the LED lamp caused a strong increase of glare in the darkest patches, as expected.

In addition to this consideration, it must be noticed that, also in this case, the increase of GE could be observed also in the condition ‘D1’, where the grayscale was toward the borders (e.g., distant from the lamp), so the glare spread to a varying extent in all the regions of the image.

The values of Glare Effect in the IR range are reported in Table 9. Additionally in this case the data trend was coherent with the visible range, and the highest GE values were registered for images ‘N17’ and ‘N18’, where the CCs grayscale was placed near the LED lamp. In this case, in ‘N18’ the black patches reached values of 5.05 and 6.34.

Now, considering the setup 3, the Glare Effect in the visible range ($GE_{400,700}^{\left(1\right)}$) has been measured using $RS^{1}$ as reference:

**Table 10 sensors-20-04374-t010:** $GE_{400,700}^{\left(1\right)}$.

HSI Number	w	g3	g2	g1	g0	bk
N22S3B0D1-T21-CCA	1.00	1.02	1.03	1.07	1.17	1.49
N22S3B0D1-T21-CCB	1.00	1.05	1.12	1.04	1.14	1.48
N23S3B0D1-T11-CCA	1.00	1.03	1.04	1.07	1.16	1.50
N23S3B0D1-T11-CCB	1.00	1.03	1.09	1.04	1.14	1.44

To make the table (of the glare effect in each acquisition) easier to read we highlighted in gray the values between 1 and 1.19, in light blue the values between 1.20 and 1.79.

and the Glare Effect in the IR range ($GE_{700,1000}^{\left(2\right)}$) has been measured using $RS^{2}$ as reference:

**Table 11 sensors-20-04374-t011:** $GE_{700,1000}^{\left(2\right)}$.

HSI Number	w	g3	g2	g1	g0	bk
N22S3B0D1-T21-CCA	1.00	0.93	0.86	1.04	1.16	1.89
N22S3B0D1-T21-CCB	1.00	0.93	0.92	0.92	1.09	1.78
N23S3B0D1-T11-CCA	1.00	0.93	0.87	1.04	1.16	1.92
N23S3B0D1-T11-CCB	1.00	0.91	0.90	0.93	1.09	1.74

To make the table (of the glare effect in each acquisition) easier to read we highlighted in gray the values between 1 and 1.19, in light blue the values between 1.20 and 1.79, yellow the values between 1.80 and 2.39.

As reported in Table 10 and Table 11, in setup 3 two exposure times were used, ‘T21’ and ‘T11’, which produced almost identical results. Additionally in this setup, the Glare Effect grew along the grayscale and the black patches were the ones that reported the highest values. Thanks to these measures we could compare the values of setup 2 with the images obtained in the same conditions but without the LED lamp and determined that glare was not caused just by the light source but remained also in this condition. More specifically, considering the results in the visible range reported in Table 10, it was possible to see that glare was present also without a white background or an intense light source in the image (as seen in setup1 with ‘B0’). In fact, the presence of a white patch was sufficient to affect with glare all the image, causing an increase of the black reflectance of 1.50 circa.

### 4.3. Glare Spectral Analysis

In this Section we present a first outlook at the glare spectral trend. The plots in Figure 10 and Figure 11 represents $GE^{\left(1\right)}$ values for the wavelengths in the visible range (Figure 10a and Figure 11a) and the $GE^{\left(2\right)}$ values in the IR range (Figure 10b and Figure 11b), with a step of 10 nm. In each plot are reported the GE values for each grayscale patch.

Considering Figure 10, the Glare Effect was computed from the data coming from the ColorChecker B in the first setup (‘S1’), with black background (‘B0’) and with the grayscale toward the center of the hyperspectral image (‘D0’). Considering the visible range (Figure 10a), the values of $GE^{\left(1\right)}$ were higher in short wavelengths (blue-violet portion of the visible spectrum), especially in the ‘bk’ and ‘g0’ patches, and its trend was statistically comparable in all the other wavelengths. Considering the $GE^{\left(2\right)}$ values in the IR range (Figure 10b), the Glare Effect increased significantly in the portion of the spectrum from 900 to 1000 nm and ‘bk’ was significantly higher in intensity than the other patches. Nevertheless, the mismatch between the acquired spectra and $RS_{2}$ affected significantly the reliability of the outcomes, which resulted in the decrease of many GE values under 1 (e.g., patches ‘g3’ and ‘g2’ in Figure 10b). In Figure 11 the spectral Glare Effect trend was analyzed for the CCB, acquired in the second setup (‘S2’), with black background (‘B0’) and with the grayscale in proximity to the LED lamp in the scene (‘D0’). As in the previous case, the values of $GE^{\left(1\right)}$ were higher for shorter wavelengths, but in this condition, the spectral trend was less smooth and presented a peak around 580 nm (in correspondence with the first peak of the raw data, Figure 7b). Considering the $GE^{\left(2\right)}$ values in the IR range, also in this case an increase of GE values was registered over 900 nm. In both the considered conditions, the glare trend was similar, with a growth in the shortest and in the longest wavelengths. Furthermore, it was clearly visible that the presence of the LED in the second setup, increased all the GE values in intensity, but the general shape was similar. An interesting difference between the two setups was reported in the patch ‘g2’, which overcame in intensity the patch ‘g1’ in ‘N14-S2’ also if it was lighter, and seemed more affected by the increase in intensity at the limits of the spectral range (below 470 nm and over 900 nm). From the glare spectral analysis, we can deduce that the Glare Effect presented some differences in specific wavelengths regions, but this could be caused by different factors like mismatches between the reference and the acquired spectra, or approximation errors during the normalization procedures. Thus, the analysis of glare spectral trend deserves further attention and research.

## 5. Dealing with Glare

Here we report a comment from one of the reviewers, since it is indicative of possible misunderstanding about glare: “It is already well known that light source uniformity (including intense bright lights) and measurement geometry, among other factors, can have an impact on the digital levels recorded in hyperspectral imaging systems (and also color ones) and so, on the reflectances calculated”.

The comment is true, however a possible misunderstanding can arise from the fact that the phrase narrows the description to particular situations or lighting setups in which high spatial differences in light distribution in the scene can cause problems (e.g., sensor pixel saturation, exposure failure) and change the recorded values. These problems can usually be solved, or controlled, modifying the scene lighting and setup.

Another source of possible confusion is considering glare as a type of camera noise like e.g., dark current sensor noise or others. Technically speaking, glare is a noise, since it produces an undesired modification in the acquired value, but it can be orders of magnitude higher. Considering the presented results, the glare magnitude on non-negligible portions of the acquired images is way bigger than the effect of a non-uniform illumination, or other sources of noise.

Glare is often not detected and considered because usually the many form of noise are measured pointwise, not as a contrast. This takes us to the goal of this paper: let the reader be aware of the effect of glare, so as to consider it in the utilization of the acquired values. We are well aware of many possible interfering effects that can influence the acquisition of reflectance values at various wavelengths. We also know which are common correction/compensation strategies to calibrate the HSI acquisition. Glare is usually on top of these all, and in some situations, it can become prevalent.

The aim of the presented tests is to check the effect of glare in different realistic acquisition setups. For this reason, we designed three different test setups in normal acquisition conditions, considering different geometries, light sources and backgrounds. We have found the effect of glare in all setups, also in images without intense light sources or white background, reporting an increment of reflectance, especially on the darkest pixels. Even in presence of other factors, like light source uneven spatial distribution, the glare effect can prevail.

What emerges to be relevant in our study, is that glare does not produce effects only when a direct light source is hitting the acquisition system, but it can generate unwanted changes in the acquired values also in setups with much more limited dynamic range. The spatial dependency of glare makes it impossible to estimate limits (or thresholds) in advance, without precise knowledge of the scene to acquire (that is usually the goal of the acquisition, not its starting point) [18,21]. In fact, each pixel, at the same time is a source of glare and is hit by the glare arriving from all the other pixels.

The alert reported in the paper regards the fact that in the hyperspectral imaging field, acquired data are considered as quantitative spectral measures. The common idea is that, after careful, calibrations (see e.g., [22]), the acquired spectral values at each point will be reasonably correct. In this consideration, the systematic effect of glare is not considered. We do not want to negate the validity of taking and exploiting spectral measures by HSI systems. However, we believe it is important to increase awareness on the glare phenomenon since, unfortunately, it affects the acquired measurement and it requires further research efforts for possible compensation strategies.

With this paper we present a way to measure this phenomenon. Dealing with it depends on the goal of the measurements and, depending on the applications, it would be appropriate, for example, to modify the acquisition setup or take into account glare in the design of the data analysis. For example, even without explicit compensation, modern data driven approaches (machine and deep learning) can learn to compensate for some effects. More explicit compensation strategies could be inspired by works in the field of Remote Sensing that aims to compensate for other (atmospherical) scattering effects [23].

## 6. Conclusions

Glare is a phenomenon that affects every image acquisition system which use lenses. It causes an unwanted light-spread on the imaging sensors and a consequent decrease in contrast and dynamic range. In this work we investigated the glare influences in hyperspectral acquisitions, characterizing its presence and its spatial variation and measuring its contribution in the final image.

In this paper, we presented a measure named Glare Effect (GE) to quantify the amount of glare in some hyperspectral images. To obtain this measure we compared the values of reflectance of the grayscale patches of a ColorChecker, measured with a contact spectrophotometer, to the spectra obtained with an HSI camera to quantify the variations introduced by the acquisition system optics and characterize glare in the final images. Different acquisition set up have been tested in order to characterize the glare spatial variation under different illumination and background conditions.

We have found that glare affected all the analyzed patches in all the tested images, adding an unwanted light contribution to the reflectance spectra, which is higher in the darker pixels. The analysis of glare spatial variation reported that Glare Effect increases in proximity to a white patch or light background and close to an intense light source. Thanks to the wide range of reflectance acquired through HSI, we have analysed the presence of glare in the visible range (from 400 to 700 nm) and in IR (from 700 to 1000 nm) range and we reported a preliminary study of glare spectral trend.

The spatial dependency of glare makes it strongly data-dependent, with effects that can be more or less tolerated depending on the acquisition purpose and use and the field of application. The goal of this paper is to rise the awareness of glare effect on the acquired data in hyperspectral imaging domain, and to stimulate research towards its compensation.

## Figures and Tables

**Figure 1 sensors-20-04374-f001:**
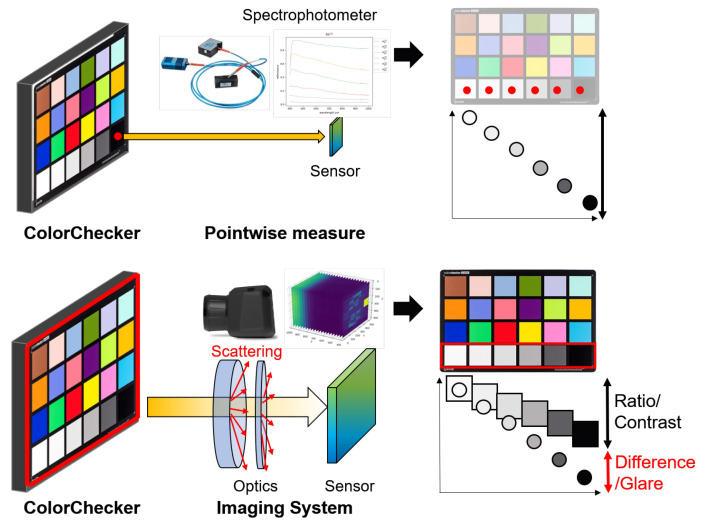
Glare effect. On the top, a graphical representation of a pointwise measure of reflectance. In this system, the reflectance spectrum is acquired directly from a portion of the surface in standard colorimetric conditions (illumination geometry, direction of observation, and light source). On the bottom, a graphical representation of an imaging system, where the scene information is acquired through a sensing pipeline which requires optics. Thus, the scattering phenomena can induce the glare effect and decrease the contrast of the final image. In this work, to provide a measure of the glare effect, we performed a comparison between the ratio of the grayscale patches acquired through a pointwise system and the ratio of the grayscale patches acquired through a hyperspectral image (HSI) system.

**Figure 2 sensors-20-04374-f002:**
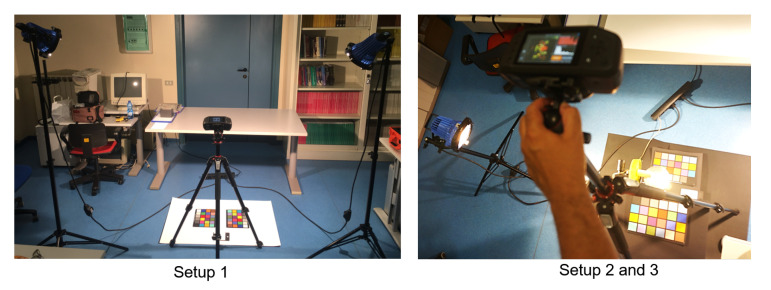
Pictures of the experimental setups.

**Figure 3 sensors-20-04374-f003:**
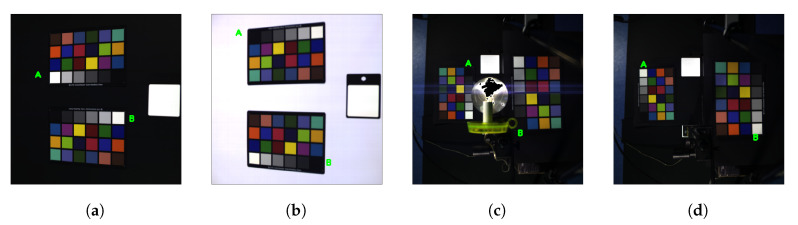
Some examples of experimental setups, in (**a**,**b**) two image from setup 1 are shown, one with dark background and the other with white background. In (**c**,**d**) one image from setup 2 and another from setup 3 are shown.

**Figure 4 sensors-20-04374-f004:**
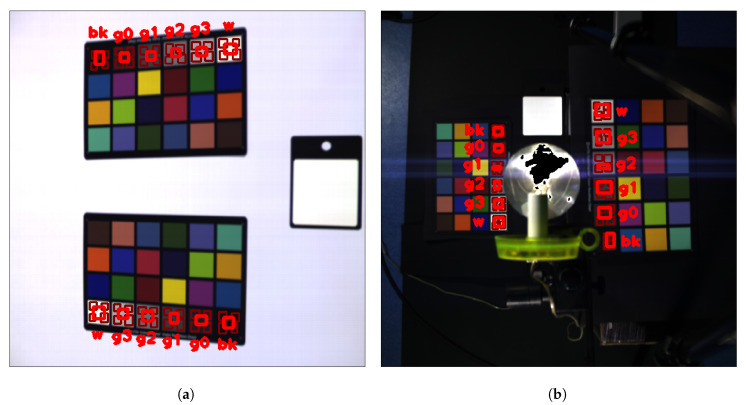
Red boxes denotes the patch regions in which spectra are collected (**a**) for image ‘N09’ and (**b**) for image ‘N17’.

**Figure 5 sensors-20-04374-f005:**
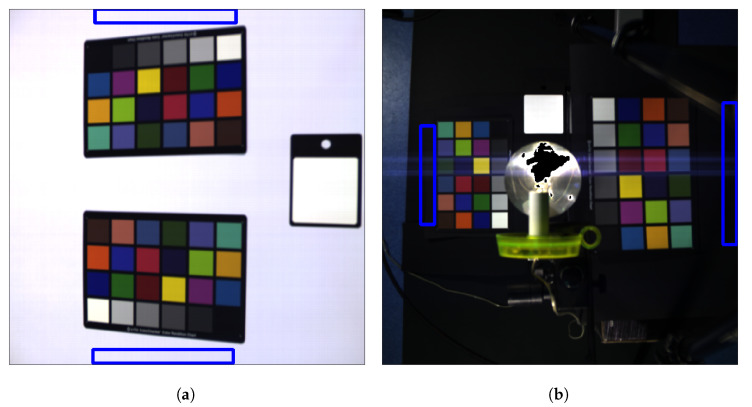
Collected background regions, (**a**) in image ‘N09’, (**b**) in image ‘N17’.

**Figure 6 sensors-20-04374-f006:**
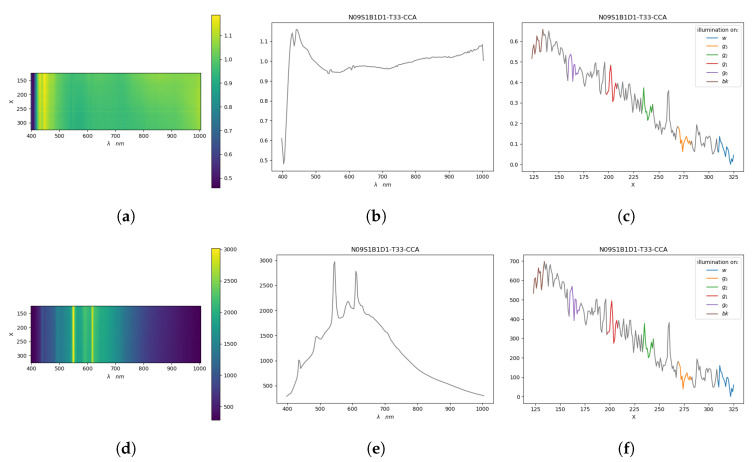
Data collected from the background regions above A ColorChecker (CCA) in N09S1B1D1T33. On the top the data of the hyperspectral data cube expressed in reflectance (**a**–**c**) and on the bottom the raw data cube values (**d**–**f**) are represented. The color maps (**a**,**d**) show the mean value of the hyperspectral data cube along the background above the CC. In (**b**,**e**) the means over both the spatial components are represented. In (**c**,**f**) the differences between energy and its minimum associated to each row of (**a**,**d**) are reported.

**Figure 7 sensors-20-04374-f007:**
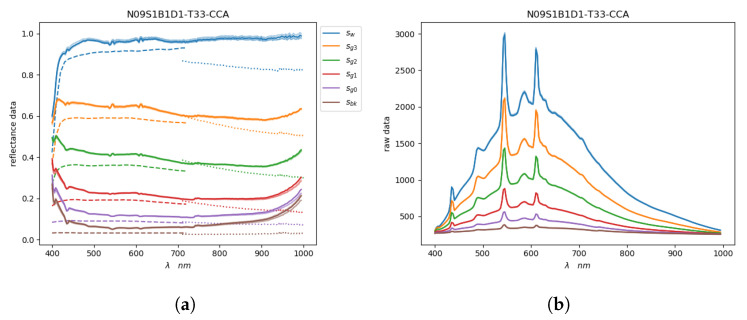
Spectra extracted from the CCA grayscale on the image N09S1B1D1T33. In (**a**,**b**) the solid lines represent the mean reflectance and raw acquired values respectively. In (**a**) the dashed lines represent the reference *RS*^(1)^ in 400 nm to 700 nm and the dotted lines represent the reference *RS*^(2)^ in 700 nm to 1000 nm.

**Figure 8 sensors-20-04374-f008:**
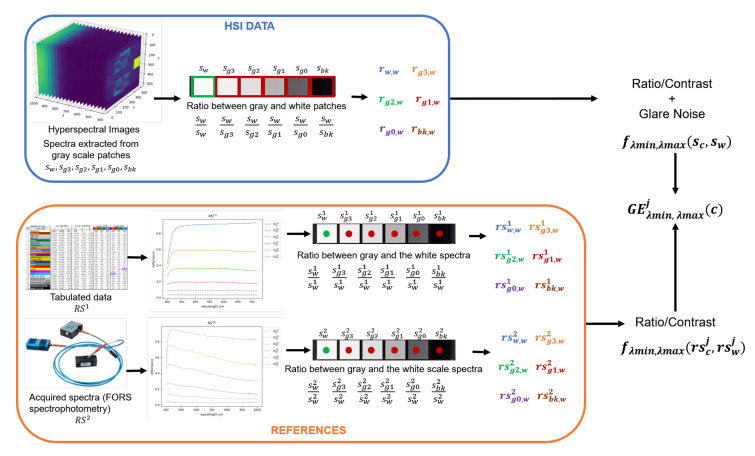
Graphical explanation of the Glare Evidence (GE) formula. The numerator is the ratio between the gray and white patches of the HSI system, which includes the glare effect introduced by the lenses in the imaging system. The denominator is the ratio between the gray and white spectra obtained from the two references (pointwise measuring systems). Thus, GE measures the noise introduced by the imaging system which is represented by the difference between the two computed ratios.

**Figure 9 sensors-20-04374-f009:**
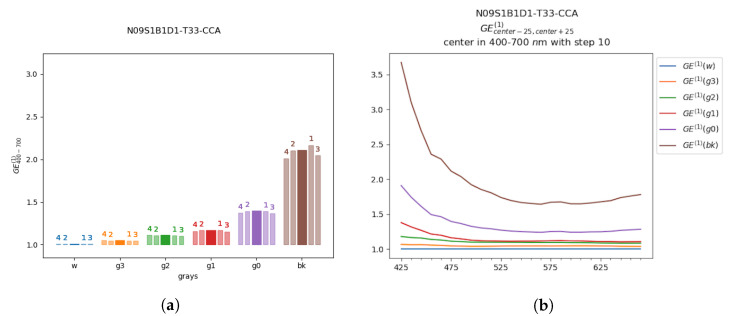
Example of *GE*^(1)^ plot representation of the CCA on image *N09S1B1D1-T33*. Plot (**a**) represents average values of glare effect of each grayscale patch in a range from 400 nm to 700 nm. The central solid color bar represents the GE values obtained from the central region of the grayscale patch; the labels 1, 2, 3, 4 refers to the GE values obtained from data of the regions toward the patch corners (see patch regions in Figure 4). In (**b**) the average values of *GE*^(1)^ are represented for each wavelength from 400 nm to 700 nm and smoothed by applying a sliding window in the considered interval.

**Figure 10 sensors-20-04374-f010:**
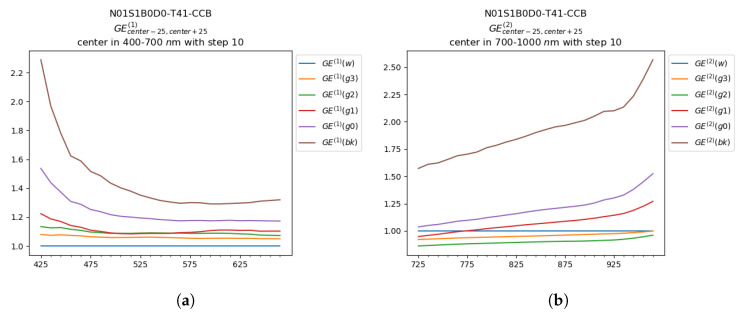
*GE*^(1)^ of the CCB of image *N01S1B0D0-T41* in (**a**) range 400 nm to 700 nm and (**b**) in the interval 700 nm to 1000 nm, both evaluated for each wavelength and smoothed by applying a sliding window.

**Figure 11 sensors-20-04374-f011:**
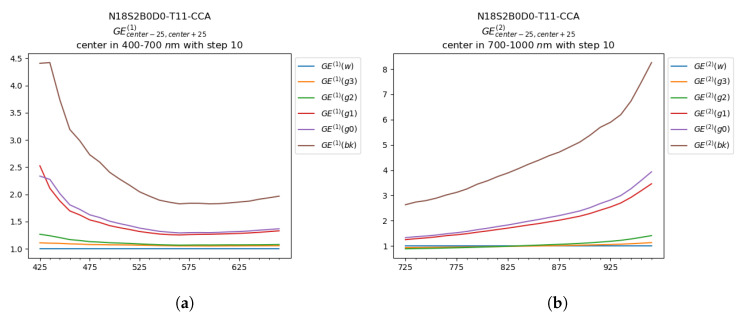
*GE*^(2)^ of the CCA of image *N18S2B0D0-T11* in (**a**) range 400 nm to 700 nm and (**b**) in the interval 700 nm to 1000 nm, both evaluated for each wavelength and smoothed by applying a sliding window.

**Table 1 sensors-20-04374-t001:** Color Name Notation.

Official Name	Used Name
white 9.5 (0.05D)	*w*
neutral 8 (0.23D)	*g3*
neutral 6.5 (0.44D)	*g2*
neutral 5 (0.70D)	*g1*
neutral 3.5 (1.05D)	*g0*
black 2 (1.5D)	*bk*

**Table 2 sensors-20-04374-t002:** Names are coded using the pattern N·S·B·D·-T· where S identifies the setup (1), B the background (black, white), D the position of the gray-scale rows (toward center or toward border) and T the integration time.

HSI Number	Associated Name
*01*	N01S1B0D0-T41
*02*	N02S1B0D1-T41
*03*	N03S1B0D0-T33
*04*	N04S1B0D1-T33
*05*	N05S1B0D0-T14
*06*	N06S1B0D1-T14
*07*	N07S1B1D1-T41
*08*	N08S1B1D0-T41
*09*	N09S1B1D1-T33
*10*	N10S1B1D0-T33
*11*	N11S1B1D1-T14
*12*	N12S1B1D0-T14

**Table 3 sensors-20-04374-t003:** Names are coded using the pattern N·S·B·D·-T· where S identifies the setup (2), B the background (black, white), D the position of the gray-scale rows (toward center or toward border) and T the integration time. The star points at the images which presents saturated white in the A ColorChecker (red) and in the B ColorChecker (blue).

HSI Number	Associated Name
*13*	N13S2B0D1-T44 **
*14*	N14S2B0D0-T42 *
*15*	N15S2B0D0-T42 **
*16*	N16S2B0D0-T42 **
*17*	N17S2B0D0-T11
*18*	N18S2B0D0-T11
*19*	N19S2B0D1-T11
*20*	N20S2B0D1-T11
*21*	N21S2B0D1-T11

**Table 4 sensors-20-04374-t004:** Names are coded using the pattern N·S·B·D·-T· where S identifies the setup (3), B the background (black, white), D the position of the gray-scale rows (toward center or toward border) and T the integration time.

HSI Number	Associated Name
*22*	N22S3B0D1-T21
*23*	N23S3B0D1-T11

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
