# Peer review of "Spatial–Spectral Evidence of Glare Influence on Hyperspectral Acquisitions"

_sensors, 2020, doi:10.3390/s20164374_

Round 1
Reviewer 1 Report
In this work, the authors investigate what they call the “glare” influence on hyperspectral acquisitions, characterizing its presence and spatial variation and measuring its contribution in the final image. They also propose a new metric called Glare Effect (GE) to quantify the effect of “glare” in some hyperspectral images, which is based on a contrast computation.
It is already well known that light source uniformity (including intense bright lights), measurement geometry, among other factors, can have an impact on the digital levels recorded in hyperspectral imaging systems (and also color ones) and so, on the reflectances calculated. And the authors demonstrate this.
The authors talk about “glare” to include all these factors, which I think might be confusing…. It can be thought that they are talking about glare caused by the optics of the acquisition system itself rather than all the other factors, which, I am sure, affect more the results in the experiments proposed.
The most relevant conclusion of the work is that different geometries and setups with different light sources, as well as different backgrounds, influence the digital records in a different manner, especially in dark colors and when intense light sources are near the patch analyzed. This is something that it is already known, also in color measurement systems, although the authors carry out a correct methodology to measure it, and propose a new parameter (GE) to account for this effect. However, in my opinion the study lacks novelty, with the exception of the GE parameter, which could be useful for characterizing these effects in new hyperspectral setups. The authors should give much more information on the effect of all factors on the readings, and give ranges (for instance, in terms of GE) in which the measurements are still useful from a spectroscopic point of view, even influenced by glare. In this sense, no practical conclusions and advices are given to work with hyperspectral systems in the present form of the manuscript. So that there is much room for improvement of the analysis of the study.
GENERAL COMMENTS
- I suggest to discuss in more detail which are the factors that influence what the authors call “glare”, along all manuscript (introduction, discussion, conclusions).
- It would be useful to describe/discuss in more detail all these factors, describe accordingly why the different setups are needed, and give more solid and useful conclusions of the experiments done (for instance in terms of QE, the ranges that are acceptable for spectroscopic measurements, the specific effect of each factor influencing the hyperspectral readings, etc.). Maybe some kind of statistics would help in this analysis.
SPECIFIC COMMENTS
- Change “push-brum” to “push-broom”.
- Page 3. Line 71. Why these 3 wavelengths are chosen? Explain better.
- In order to remove the influence of the non-uniformities in the illumination, why the authors does not use a greater diffusing white reference covering the whole FOV, in terms of x and y (and not only averaging with the wavelength, WR(x,y,l) and not WR(l))? In this way, they could account only for the glare due to intense/bright light sources near the patches.
- Page 4. Lines 102-103. What about the influence of the geometry in the spectroscopic measurements with the hyperspectral system? And what about the geometry used in the reference measurements? Why 45º?
- Page 4. Line 106. Change “for the purpose of limit” to “for the purpose of limiting”
- Explain better why the need of the different setups used? It is not clear to me. Why you need all them? Or whay not others?
- Page 8. Figure 6 caption. Change (d,e,d) for (d,e,f).
- Page 9. Linea 164-165. Change “to better to highlight” for “to better highlight”.
- Page 9. Line 175. Change “Thank this” to “Thanks to this”
- Page 11. Line 201. Review sentence grammar (order).
- Page 12. Line 238. Review sentence grammar (order).
- Page 15. Line 299. There is a “.” In the beginning of the line. Please remove.
- Page 16. Line 310. Remove “s” at the end of “every image acquisition systems”
Author Response
We would like to thank the Associate Editor and the Reviewers for their useful and detailed comments. We addressed all of them in the revised version of the manuscript, and we believe the overall paper quality has improved thanks to their contributions. Answers to all Reviewers’ points follow below.
_____________________________________________________________
In this work, the authors investigate what they call the “glare” influence on hyperspectral acquisitions, characterizing its presence and spatial variation and measuring its contribution in the final image. They also propose a new metric called Glare Effect (GE) to quantify the effect of “glare” in some hyperspectral images, which is based on a contrast computation.
This clear description offers us the chance to underline the point that usually the many form of noise are measured pointwise, not as a contrast. For this reason, the high effect of glare is often not detected and considered.
There also could be some linguistic overlap of meaning that might lead to confusion. For this reason, in the first parts of the introduction we added a clarification sentence to further disambiguate and prevent misunderstandings.
It is already well known that light source uniformity (including intense bright lights), measurement geometry, among other factors, can have an impact on the digital levels recorded in hyperspectral imaging systems (and also color ones) and so, on the reflectance calculated. And the authors demonstrate this.
Well aware of interfering effects that influence reflectance values at various wavelengths which are commonly the target of correction/compensation strategies, our main goal in this work is to talk about an additional effect that is usually neglected, that, in some situations, it can become prevalent.
To better present/prevent possible misunderstanding in a discussion including goals and results of the paper, we have added a discussion section (number 5) before the conclusions.
Our aim is to check if the effect of glare is present in every possible acquisition setup. For this reason, we put ourselves in a normal acquisition condition and introduced a measuring parameter which is specific to glare assessment, to demonstrate that, even in presence of confounding and only partially compensated factors (such as light source unevenness), the glare effect can prevail, with consequences on the image contrast that cannot be explained with other effects that are usually taken into account. Moreover, what emerges to be relevant in our study, and calling for disambiguation about the meaning and consequences of the “glare” conceptualization, is that glare does not emerge and produce effects only when a direct light source is glaring the acquisition system (whether it is the human eye or a camera) but it can generate unwanted changes in the acquired values, that depend on the arrangement of bright areas in the scene.
The alert reported in the paper regards the fact that, in the hyperspectral imaging field, reflectance values obtained after proper scene and camera calibration are often considered (more than in color imaging) as correct quantitative spectral measures. The common idea is that, after careful calibrations, the acquired values at each point will be reasonably correct. In this consideration, the systematic effect of glare is not considered. We do not want to negate the validity of taking and exploiting spectral measures by HSI systems. However, we believe it is important to create or increase awareness on the glare phenomenon since, unfortunately, it affects the acquired measurement and would require some research efforts in possible compensation strategies.
We included the above observations in the discussion section.
The authors talk about “glare” to include all these factors, which I think might be confusing…. It can be thought that they are talking about glare caused by the optics of the acquisition system itself rather than all the other factors, which, I am sure, affect more the results in the experiments proposed.
The belief of the reviewer is perfectly understandable and is shared by the majority of experts in the field. Unfortunately, the results presented in this paper depict another reality, as well as many other experiments run on standard color images. Before the tests, we had this belief we too. When we first realized that glare unwanted changes were so high, it was a real surprise. From this derives the aim of the paper. Since it comes from measures, the extent of the glare effect is not a matter of opinion.
Indeed, we address “glare” in its physical definition (actually related to the optics of the acquisition system) and its possible prevalence with respect to other sources of measuring error. Therefore, if we correctly interpret the final part of the reviewer sentence, the “glare” in its physical definition (and not in a general all-inclusive sense) can instead affect results more than other causes in some specific but not infrequent conditions (darker regions contrast, proximity of bright patches).
We included the above observations in the discussion section.
The most relevant conclusion of the work is that different geometries and setups with different light sources, as well as different backgrounds, influence the digital records in a different manner, especially in dark colors and when intense light sources are near the patch analyzed. This is something that it is already known, also in color measurement systems, although the authors carry out a correct methodology to measure it, and propose a new parameter (GE) to account for this effect. However, in my opinion the study lacks novelty, with the exception of the GE parameter, which could be useful for characterizing these effects in new hyperspectral setups.
To reply to this point we need to continue on the path of the previous points.
We are glad the reviewer points out the novelty of GE way to measure glare effect. This is exactly the intended novelty of the paper. If someone considers only the pointwise values, he/she will never detect the systematic effect of lens glare. It is necessary to consider contrast, or more simply relative values. The simple assumption that relationship (ratios) in the scene should be acquired preserving the original ratios is no more valid due to glare.
We are aware we are not discovering a new phenomenon, but we have presented this work because the physical phenomenon of glare is not considered (or in some case misunderstood) in both color and HSI fields. This is demonstrated by the fact that glare is not comprised in models and strategies that are currently implemented and exploited for HSI calibration and compensation.
The reported tests confirm this. As far as we know, this is the first work pointing out the problem of glare in HSI on common acquisition setups and trying to highlight that this can be a significant source of error, and that differs from those that are normally considered and compensated. Here lies the novelty of work, together with the proposal of a way to measure it.
The authors should give much more information on the effect of all factors on the readings, and give ranges (for instance, in terms of GE) in which the measurements are still useful from a spectroscopic point of view, even influenced by glare. In this sense, no practical conclusions and advices are given to work with hyperspectral systems in the present form of the manuscript. So that there is much room for improvement of the analysis of the study.
The relevance and insidiousness of the glare can be different depending on the object scene spatial arrangement and on the illumination setup. Our goal is to present this and to offer a way to measure this phenomenon. Depending on the applications, it would be appropriate, for example, to modify the acquisition setup or take into account the phenomenon in the design of the data analysis. For example, even without explicit compensation, modern data driven approaches (machine and deep learning) can learn to compensate for some effects.
The spatial dependency of glare makes impossible to estimate limits in advance, without knowledge of the scene to acquire. Glare mainly influences values in dark shades and in region closer to bright areas.
For the above reason, without an exact knowledge of the scene to acquire, glare cannot be removed. This strong statement comes from an ISO standard and other studies reported in the paper. However, the interest on this problem is recently increasing and this work wants to be also a stimulus to research aimed at dealing with the glare effects.
With this contribution we also want to stimulate research in advanced compensation strategies.
All these reasoning and observations about GE entities and possible consequences are included in the new discussion section, and we thank again the reviewer for stimulating us in doing this.
GENERAL COMMENTS
1. I suggest to discuss in more detail which are the factors that influence what the authors call “glare”, along all manuscript (introduction, discussion, conclusions).
We did it according to what exposed above
2.It would be useful to describe/discuss in more detail all these factors, describe accordingly why the different setups are needed, and give more solid and useful conclusions of the experiments done (for instance in terms of QE, the ranges that are acceptable for spectroscopic measurements, the specific effect of each factor influencing the hyperspectral readings, etc.). Maybe some kind of statistics would help in this analysis.
We did it according to what exposed above, in particular, we discussed in more details all factors related to the use of glare measurements. We avoided to prescribe acceptability limits since they strongly depend on the application and on the intended use of HSI acquisitions.
SPECIFIC COMMENTS
1.Change “push-brum” to “push-broom”.
Fixed, thank you
2.Page 3. Line 71. Why these 3 wavelengths are chosen? Explain better.
We specified that these wavelengths are directly from the Specim IQ camera software for true color image displaying
3.In order to remove the influence of the non-uniformities in the illumination, why the authors does not use a greater diffusing white reference covering the whole FOV, in terms of x and y (and not only averaging with the wavelength, WR(x,y,l) and not WR(l))? In this way, they could account only for the glare due to intense/bright light sources near the patches.
We fully understand the concern and with our approach we wanted to take into account and give an answer to what you are kindly pointing out. However, we did not want to work into unrealistic conditions, thus we followed standard acquisition procedures that cannot assume to have white reference panels covering the whole scene (especially in large field or open-air acquisitions). We was aware that illumination non-uniformity could have been confused with glare but we monitored the degree of non-uniformity (by measures on the white paper background panel). Measures demonstrate that glare can widely overcome the light non-uniformity expected contribution without the need to strictly separate the two possibly confounding effects (therefore without having to counteract light non-uniformity in unpractical ways).
4. Page 4. Lines 102-103. What about the influence of the geometry in the spectroscopic measurements with the hyperspectral system? And what about the geometry used in the reference measurements? Why 45º?
The spectroscopic measures are point based therefore this does not introduce light angle compatibility issues between the two acquisition methods, devices and setups. The 45º angle is an inbuilt parameter of the spectroscopy device. For matte surfaces like the color checker it is a standard way to acquire reflectance.
5.Page 4. Line 106. Change “for the purpose of limit” to “for the purpose of limiting”
Fixed, thank you
6.Explain better why the need of the different setups used? It is not clear to me. Why you need all them? Or whay not others?
Thank you for pointing out. The different setups gave us the possibility to observe and discuss different aspects and sources of glare in different acquisition contexts (various configurations in setup 1) and to assess the presence of glare with and without direct light into the scene (setup 2 vs 3).
Better explanations have been given in section 2.3
8.Page 8. Figure 6 caption. Change (d,e,d) for (d,e,f).
9.Page 9. Linea 164-165. Change “to better to highlight” for “to better highlight”.
10.Page 9. Line 175. Change “Thank this” to “Thanks to this”
11.Page 11. Line 201. Review sentence grammar (order).
12.Page 12. Line 238. Review sentence grammar (order).
13.Page 15. Line 299. There is a “.” In the beginning of the line. Please remove.
14.Page 16. Line 310. Remove “s” at the end of “every image acquisition systems”
All fixed, thank you
Reviewer 2 Report
This is paper has an original contribution to knowledge, and I recommend it for publication after a revision. I will do my best to give constructive comments to improve the final version of this paper;
For the introduction, it is essential to highlight the significance of your work and how it will benefit other researchers in the area.
Review all references as they are not included in the version.
Your paper will benefit from increasing the literature review of other related work on Glare.
Add the limitations of this study (for example; all these tests were conducted in a controlled experimental setup, further work will still be needed for validating this method on field studies).
Use the last section of your discussion to communicate the applications of your method, and how it can be further developed into a calibration model for this type of data acquisition. For example (your work can be very beneficial if it can be used as an additional calibration model for Glare in buildings data acquisition such as the work by :
- Pierson, A. Jacobs, J. Wienold, M. Bodart, Luminance maps from High Dynamic Range imaging: photometric, radiometric and geometric calibrations, in: Lux Europa 2017, 2017.
- Wagdy, V. Garcia-Hansen, G. Isoardi, K. Pham, A parametric method for remapping and calibrating fisheye images for glare analysis, Buildings, 9 (10) (2019) 219.
- Hansen, M. Andersen, J. Wienold, HDR IMAGES FOR GLARE EVALUATION: COMPARISON BETWEEN DSLR CAMERAS, AN ABSOLUTE CALIBRATED LUMINANCE CAMERA AND A SPOT LUMINANCE METER, 2018.
- Y. Suk, M. Schiler, K. Kensek, Investigation of existing discomfort glare indices using human subject study data, Building and Environment, (2017).
Expand on the current limitation and the future work, and the potential of transforming this method into a calibration model.
Author Response
We would like to thank the Associate Editor and the Reviewers for their useful and detailed comments. We addressed all of them in the revised version of the manuscript, and we believe the overall paper quality has improved thanks to their contributions. Answers to all Reviewers’ points follow below.
_______________________________________________________________
This is paper has an original contribution to knowledge, and I recommend it for publication after a revision. I will do my best to give constructive comments to improve the final version of this paper;
For the introduction, it is essential to highlight the significance of your work and how it will benefit other researchers in the area.
We have added more explanations in the introduction and a section (number 5) entirely dedicated to this.
Review all references as they are not included in the version.
We are not sure to have fully understood this point. However, references have been checked and updated on some topic.
Your paper will benefit from increasing the literature review of other related work on Glare.
Thank you for the suggestion, we added few more references on glare physical phenomenon.
Add the limitations of this study (for example; all these tests were conducted in a controlled experimental setup, further work will still be needed for validating this method on field studies).
Despite our goal was to reproduce quite realistic acquisition conditions we added a sentence related to what suggested in the new discussion section.
Use the last section of your discussion to communicate the applications of your method, and how it can be further developed into a calibration model for this type of data acquisition. For example (your work can be very beneficial if it can be used as an additional calibration model for Glare in buildings data acquisition such as the work by :
- Pierson, A. Jacobs, J. Wienold, M. Bodart, Luminance maps from High Dynamic Range imaging: photometric, radiometric and geometric calibrations, in: Lux Europa 2017, 2017.
- Wagdy, V. Garcia-Hansen, G. Isoardi, K. Pham, A parametric method for remapping and calibrating fisheye images for glare analysis, Buildings, 9 (10) (2019) 219.
- Hansen, M. Andersen, J. Wienold, HDR IMAGES FOR GLARE EVALUATION: COMPARISON BETWEEN DSLR CAMERAS, AN ABSOLUTE CALIBRATED LUMINANCE CAMERA AND A SPOT LUMINANCE METER, 2018.
- Suk, M. Schiler, K. Kensek, Investigation of existing discomfort glare indices using human subject study data, Building and Environment, (2017).
Expand on the current limitation and the future work, and the potential of transforming this method into a calibration model.
Thanks for this suggestion. We found useful to add some of these references, they help us to describe how the term glare refers to both the effect and the cause of the same phenomenon. This will help to describe the complexity of it and hopefully to prevent potential misunderstanding. Current research is far from a general-purpose calibration method to deal with glare. We agree with the reviewer that having such a method would be very beneficial. This contribution aims at increasing awareness about how glare can affect HSI measurements by introducing methods to assess this phenomenon. We also aim at stimulating investigation in the direction of finding novel compensation strategies. We convey all these aspects with further observations in the new last section of the paper.
Round 2
Reviewer 1 Report
There are few minor spell errors in the new parts included in the text.